# Gastroduodenal Stenting with a Flexible Stent Demonstrates Favorable Clinical Effectiveness despite Gradual Expansion: A Multicenter Prospective Study

**DOI:** 10.3390/jcm12030850

**Published:** 2023-01-20

**Authors:** Hiroaki Shigoka, Masao Toki, Sho Takahashi, Naminatsu Takahara, Katsuya Kitamura, Eisuke Iwasaki, Kazunari Nakahara, Hiroyuki Isayama, Yousuke Nakai, Iruru Maetani

**Affiliations:** 1Division of Gastroenterology and Hepatology, Department of Internal Medicine, Toho University Ohashi Medical Center, Tokyo 153-8515, Japan; 2Department of Gastroenterology and Hepatology, Kyorin University School of Medicine, Tokyo 181-8611, Japan; 3Department of Gastroenterology, Graduate School of Medicine, Juntendo University, Tokyo 113-8421, Japan; 4Department of Gastroenterology, Graduate School of Medicine, The University of Tokyo, Tokyo 113-0033, Japan; 5Department of Gastroenterology and Hepatology, Tokyo Medical University Hachioji Medical Center, Tokyo 193-0998, Japan; 6Division of Gastroenterology, Department of Medicine, Showa University School of Medicine, Tokyo 142-8555, Japan; 7Division of Gastroenterology and Hepatology, Department of Internal Medicine, Keio University School of Medicine, Tokyo 160-8582, Japan; 8Department of Gastroenterology and Hepatology, St. Marianna University School of Medicine, Kawasaki 216-8511, Japan; 9Department of Endoscopy and Endoscopic Surgery, The University of Tokyo Hospital, Tokyo 113-8655, Japan

**Keywords:** gastroduodenal stent, gastroduodenal obstruction, gastric outlet obstruction

## Abstract

Aims: This study aimed to evaluate the effectiveness and safety of stenting with a flexible braided self-expandable metal stent (SEMS) for unresectable malignant gastric outlet obstruction (GOO). Methods: Palliative stenting was prospectively carried out at seven university hospitals between October 2017 and August 2020. All procedures were performed using a flexible branded SEMS of the same brand. The primary endpoint was clinical success rate at 7 days after stenting. Secondary endpoints were procedural success rate, adverse events, recurrent gastric outlet obstruction (RGOO), and patient survival time. Results: Sixty patients were enrolled. The procedural and clinical success rates were 100% and 90%, respectively. RGOO occurred in 15 cases (25%). Adverse events other than RGOO were found in seven cases (12%). The 50% survival time was 75.5 days (range: 52–97 days). Median expansion rates at 1, 3, and 7 days after stenting were 55%, 65%, and 75%, respectively. Conclusions: A flexible braided stent woven with relatively thin wires was used for malignant GOO. Despite a gradual expansion with slightly lower expansile force, the stent functioned sufficiently well and showed favorable results. Clinical Trials Registry ID: UMIN000029496.

## 1. Introduction

Stenting for gastric outlet obstruction (GOO) is associated with a high procedural success rate (97–100%). However, the clinical success rate of this procedure ranges 84–93% [1,2,3,4,5,6,7]. Moreover, the procedure may not be effective in approximately 10% of cases. This can be attributed to conditions such as cachexia or dissemination, which are caused by decreased bowel peristalsis. In addition, severe stenosis [8] and poor stent expansion [9] have been reported; for such cases, the use of improved devices may provide effective treatment. It has also been reported that the use of a self-expandable metallic stent (SEMS) with low flexibility (high axial force) is linked to risk of stent dysfunction due to contact with the area of anatomical flexure, thereby leading to intestinal perforation. However, it has been reported that the use of SEMS with small axial force reduces this risk [10]. Of note, axial and radial forces are correlated to some extent. Hence, a stent with a small axial force tends to have a small radial force, which may complicate expansion at the site of obstruction. The aim of this study was to evaluate the effectiveness and safety of stenting with a flexible braided stent for unresectable malignant GOO.

## 2. Methods

### 2.1. Patients

Stenting was performed prospectively for GOO at seven university hospitals (Toho University Ohashi Medical Center, Kyorin University Hospital, Juntendo University Hospital, The University of Tokyo Hospital, Showa University Hospital, Keio University Hospital, and St. Marianna University Hospital) between October 2017 and August 2020. The criteria for the selection of patients were as follows: age ≥ 20 years; presence of malignant gastroduodenal obstruction; and willingness to participate in this study. The exclusion criteria were as follows: contraindications to endoscopic procedures; unsuitability of patients to participate in this study, as deemed by the attending physician or physician performing the procedure; and a Gastric Outlet Obstruction Scoring System (GOOSS) score of 3. GOOSS was used to evaluate the severity of obstructive symptoms (GOOSS 0 = no oral intake, 1 = liquids, 2 = soft solids, 3 = low residue or full diet), proposed by Adler and Baron [11]. This study was approved by the ethics committees of the respective institutions (approval number of the Ethics Committee of Toho University Ohashi Medical Center: H17020, approved by ethics review on 21 August 2017). This study was registered in the University Hospital Medical Information Network (UMIN) Clinical Trials Registry system on 11 October 2017 (ID: UMIN000029496). All patients enrolled in this study provided written informed consent. This study was performed in accordance with the ethical principles of the 1964 Declaration of Helsinki. Patient data were recorded in a secure electronic data capture system on the Internet with encrypted content for confidentiality purposes; access to these data is restricted.

### 2.2. Devices

All procedures were performed using the WallFlex™ Duodenal Soft stent (Boston Scientific Corporation, Marlborough, MA, USA). This is an uncovered SEMS with a braided Nitinol wire mesh (Figure 1). Compared with the conventional WallFlex™ stent (Boston Scientific Corporation, Marlborough, MA, USA), the wire diameter of this stent is approximately 20% smaller and the flexibility is approximately 30–60% greater. The stent delivery system is reduced from 10 Fr to 9 Fr. The stent diameters are 18, 20, and 22 mm, each 5 mm larger on the oral side. Stent lengths of 6, 9, and 12 cm are available. In this study, only stents with a diameter of 22 mm were used. The selection of stent length was at the discretion of the attending physician at the time of stenting.

### 2.3. Stenting Procedure

In all cases, the procedure was performed under conscious sedation. A straight-view or side-view endoscope with a forceps channel of ≥3.2 mm was advanced up to the stenosis. An endoscopic retrograde cholangiopancreatography catheter and a 0.025–0.035 inch biliary guidewire were used to cross the stenosis and perform contrast-enhanced imaging of the area. After confirming the stenosis length, the appropriate stent was selected. The stent length was selected to be approximately 3–4 cm longer than the stenosis length. Subsequently, the stent delivery system was advanced through the endoscope along the guidewire and deployed from the anal side. Finally, the procedure was completed by confirming proper placement under fluoroscopic or endoscopic guidance. All procedures were carried out by expert endoscopists in each hospital familiar with gastroduodenal stenting.

### 2.4. Follow-Up and Evaluation Parameters

In principle, patients were hospitalized for 1 week to evaluate their condition. and GOOSS was evaluated once daily. Simple abdominal X-ray images were captured at 1, 3, and 7 days after stenting, and blood tests were performed at 1 and 7 days after stenting. If the simple abdominal X-ray images showed that the stent was properly positioned and deployment was sufficient, the patients were allowed to drink water and gradually increase their level of food intake to the extent possible.

### 2.5. Study Endpoint and Parameter Definition

The primary endpoint was the clinical success rate at 7 days after stenting. Clinical success was defined as an improvement of at least 1 point from the time before stenting, as assessed by the GOOSS score at 7 days after stenting. Secondary endpoints were procedural success, adverse events, recurrent gastric outlet obstruction (RGOO), and patient survival time. Procedural success was defined as the successful placement of the stent in the intended site. Adverse events were defined as stent-related accidental symptoms. RGOO was defined as a relapse of obstructive symptoms due to stent occlusion or deviation. In addition, obstruction was defined as a GOOSS score of 0 due to stent occlusion. Migration was defined as stent migration away from the stent site. Insufficient expansion was defined as unsatisfactory expansion necessitating balloon dilation and/or additional stent placement. Stent abutment (SA) was defined as stent edge protrusion into the duodenal wall or duodenal flexure. Finally, kinking was defined as stent bending at the flexure. After stenting, prognosis was investigated for at least 1 year or until the death of the patient.

### 2.6. Statistical Analysis

For all endpoints, intention-to-treat analysis was performed. All patients, except those who were ineligible or withdrew consent, were included in the efficacy evaluation. Similarly, all patients, except those who withdrew consent, were included in the safety evaluation.

Continuous variables are presented as median values and ranges. Categorical variables are presented as counts and percentages. Survival time and time to RGOO were determined using the Kaplan–Meier method. The Friedman test and Bonferroni’s multiple comparison test were used to evaluate changes in the GOOSS score and stent expansion rates among three or more corresponding groups. The EZR Version 1.55 software (Saitama Medical Center, Jichi Medical University, Saitama, Japan) was used for statistical analysis.

## 3. Results

### 3.1. Patient Characteristics

A total of 60 patients (30 males and 30 females) were enrolled during the study period (Table 1). The median age was 74 years (range: 40–92 years). Primary diseases included pancreatic cancer (*n* = 32, 53%), gastric cancer (*n* = 16, 27%), biliary tract cancer (*n* = 7, 12%), and other diseases (*n* = 5, 8%). The stenosis sites were pylorus (*n* = 10, 17%) D1 (*n* = 7, 12%), D2 (*n* = 21, 35%), D3 (*n* = 14, 23%), D4 (*n* = 2, 3%), G-D (B1 reconstruction; *n* = 5 cases, 8%), and G–J (B2 reconstruction; *n* = 1, 2%). The GOOSS scores before duodenal stenting were 0 in 50 cases (83%), 1 in eight cases (13%), and 2 in two cases (3%).

### 3.2. Procedure of Duodenal Stenting

Details of duodenal stenting are shown in Table 2. The procedural success rate was 100% (60/60) (Table 3). The stents measured 22 mm in diameter and 6, 9, and 12 cm in length in nine (15%), 26 (43%), and 22 (37%) patients, respectively. Two stents were placed in three patients (5%): 12 cm + 9 cm in two patients (3%) and 12 cm + 6 cm in one patient (2%). Bile duct drainage was performed in 13 patients (22%) due to simultaneous obstruction of the bile duct and duodenum.

### 3.3. Transition of GOOSS and Clinical Success

The GOOSS scores gradually improved after stenting, with a median (interquartile range) of 0 (0–0), 1 (0–1), 1 (1–2), and 2 (1–3) before the procedure and at 1, 3, and 7 postoperative days, respectively (Figure 2). There were 54 cases with improvement in the GOOSS score of at least 1 point at 7 days after stenting, with a clinical success rate of 90% (Table 3). There was no improvement in six cases (10%). Among them, one patient maintained a GOOSS score of 1 and another had a score of 2. The GOOSS score remained 0 in four cases, three of which underwent balloon dilation and stent addition due to insufficient expansion. Two patients were able to begin oral intake at 2 days after balloon dilation and stent addition (at 9 and 13 days after initial stenting). The remaining patient had stent expansion after balloon dilation and stent addition; however, the patient was unable to begin oral intake, most likely due to decreased peristalsis caused by cancerous peritonitis.

### 3.4. Time to RGOO

Figure 3 shows the time to RGOO after duodenal stenting. RGOO developed in 15 patients (25%). Tumor ingrowth occurred in seven patients (12%), six of whom underwent additional duodenal stenting. One patient did not request treatment and received conservative therapy. Four patients (7%) developed obstruction due to abutment of the stent edge caused by shortening of the duodenal stent (Figure 4). All patients were able to resume food intake as a result of improvement following the additional duodenal stenting. Insufficient expansion was observed in three patients (5%), and all patients underwent balloon dilation and additional duodenal stenting. Migration was observed in two patients (3%); spontaneous stent excretion occurred in one of those patients on day 7 after deviation to the anal side. The patient’s GOOSS score also improved from 0 to 1, and repeat duodenal stenting was not requested. The other case involved insufficient expansion of the stent. Hence, a second stent was placed to overcome unsatisfactory expansion. However, the additional stent migrated into the stomach the following day. Although the migrated stent was removed endoscopically, insufficient expansion remained. Balloon dilation and additional duodenal stenting were performed after 11 days.

### 3.5. Stent Expansion Rate

Stent expansion rates up to 1 week after duodenal stenting are shown in Figure 5. The median (interquartile range) expansion rates at 1, 3, and 7 days after stenting were 55% (45–65%), 65% (58.75–75%), and 75% (67.5–80%), respectively, indicating favorable expansion in all cases.

### 3.6. Adverse Events

Adverse events other than RGOO occurred after stenting in seven patients (12%). Cholangitis occurred in three patients, in whom the stent crossed the duodenal papilla; two of those patients improved after conservative treatment without drainage. The remaining patient developed sepsis and disseminated intravascular coagulation due to severe cholangitis the day after duodenal stenting. Although percutaneous biliary drainage was performed, the patient expired due to uncontrollable sepsis and disseminated intravascular coagulation. This patient had stage 3 pancreatic cancer and stenosis of the third portion of the duodenum. Perforation occurred in two patients (3%), one of whom had gastric cancer (stage 4B, no ascites, and B2 reconstruction G-J stenosis). Moreover, the perforation in this patient occurred 104 days after duodenal stenting. Surgery was not indicated in this patient due to the presence of end-stage gastric cancer, and the patient expired due to perforative peritonitis. The other patient had pancreatic cancer (stage IV, ascites, and D1 stenosis). The patient expired 10 days later due to deterioration of the general condition, despite a blood transfusion for low hemoglobin levels following duodenal stenting. A pathological autopsy revealed perforation on the oral side of the duodenal stent. Acute cholecystitis occurred in one patient, who experienced improvement following percutaneous transhepatic gallbladder drainage. Mild acute pancreatitis occurred in one patient and resolved after conservative treatment.

### 3.7. Survival

At the end of this study (30 November 2021), all patients, except one, had expired. The survival curve after duodenal stenting is shown in Figure 6. The median survival time was 75.5 days (95% confidence interval: 52–97 days). There were 55 deaths (92%) due to primary diseases. Other causes of death included perforative peritonitis, perforation and hemorrhage, and severe cholangitis (one case each), as well as sudden death of unknown cause (one case). Chemotherapy and palliative radiation were administered after duodenal stenting in 12 patients (20%) and one patient (2%), respectively. Dendritic cell vaccine therapy was administered in one patient (2%).

## 4. Discussion

This multicenter, prospective cohort study evaluated the use of a flexible braided stent for malignant GOO. The analysis yielded favorable results, with a procedural success rate of 100% and a clinical success rate (i.e., improvement in the GOOSS score after 1 week) of 90%. The present findings are similar to those of previous reports [1,2,3,4,5,6,7].

Several systematic reviews have been published [4,5,12] reporting procedural and clinical success rates of 96–97.3% and 85.7–89%, respectively. A prospective study of 31 patients in whom the WallFlex™ Duodenal Soft stent (Boston Scientific Corporation, Marlborough, MA, USA) was used (i.e., the same stent used in this study) [7] reported procedural and clinical success rates of 97% and 87%, respectively. The studies conducted thus far have reported varied clinical success rates. Nevertheless, previous reports [1,2,13,14] using the same definition of clinical success as this study reported rates of 75–96.2%, and the results of this study compared favorably with those.

The radial force of the stent used in this study is decreased due to a reduction in the wire diameter compared with that of the original WallFlex™ stent. Hence, there were concerns regarding the effectiveness of the stent. The expansion rate was observed over time, with a median expansion rate of 75% recorded at 7 days after stenting. The clinical success rate, defined as an increase in the GOOSS score by at least 1 point, was 90%. This rate was comparable to those previously reported, suggesting that the capacity of this stent for expansion is sufficient for clinical use.

Three days after stenting, the expansion rate was 65% (median), showing gradual expansion from 55% (median) at 1 day after stenting; the GOOSS score was 75% at 3 days after the procedure. The duodenal stent expansion rates over time have not been shown in previous reports, and the appropriate stent expansion rate is unknown. As noted above, definitions of clinical success rate also vary between studies, and some reports do not clearly define the evaluation date. Although the stent used in this study may expand more slowly than stents with higher radial force, clinical success was achieved in 90% of enrolled cases. This finding suggested that a sufficient clinical effect could be expected.

There was no improvement in the GOOSS score in six patients (10%). One of these patients maintained a GOOSS score of 1 (expansion rate of 95% at day 7), while another maintained a score of 2 (expansion rate of 55% at day 7) and was able to continue oral intake. In four patients, the GOOSS score remained 0 from before to 1 week after stenting. There were three cases of poor stent expansion (expansion rates of 20%, 25%, and 50% at day 7). Balloon dilation and stent addition were performed, resulting in expansion rates of 60%, 40%, and 60%, respectively. Of those patients, two were able to initiate oral intake; in contrast, one patient was unable to commence oral intake even after stent expansion and was judged to have poor peristalsis due to cancerous peritonitis. The remaining patient exhibited a good stent expansion rate (85%); however, ascites was also observed possibly due to poor peristalsis caused by cancerous peritonitis.

Insufficient expansion (insufficient deployment) was reported for 0.4% of uncovered SEMS in the systematic review reported by van Halsema et al. [12], 0% in the randomized controlled trial conducted by Maetani et al. [15], and 1.59% in the retrospective study performed by Hori et al. [16]. This evidence suggested that the incidence of insufficient expansion may be high in this study. Nevertheless, clinical success was achieved in 90% of the cases, indicating good performance. In some patients, such as those with severe stenosis, balloon dilation or stent addition may be necessary. Nonetheless, since it is difficult to determine the ease of deployment preoperatively, it may be necessary to reach a decision based on the course of the patient after stenting. Ye et al. reported that a stent expansion rate ≥ 75% at 1 day after stenting was associated with stent patency [13]. However, in the present study, there were 52 cases with expansion rates < 75% at 1 day after stenting. Moreover, clinical success was achieved in 88% of these cases. These findings suggested that the appropriate index of expansion rate may differ depending on the type of stent used. Ye et al. [13] reported that the incidence of RGOO was 32.2%. In the present study, the incidence of RGOO was lower (15 patients; 25%).

In this study, RGOO was observed in 15 patients (25%). This rate is similar to that noted in the systematic review reported by Dormann et al. [4] (22.3%). The most common cause of RGOO was tumor ingrowth, which was present in 12% of the patients. This rate is comparable to those reported by Dormann et al. (17.2%) and for uncovered SEMS in a large-scale study conducted in Japan (12%) [14].

In this study, SA occurred in four patients. Park et al. [17] reported that SA is an accidental symptom associated with stenting for GOO. They stated that shortening of the stent causes the stent edge to come in contact with the area of anatomical flexure (e.g., the superior duodenal angle), resulting in obstruction. This suggested that SA may have been overlooked in previous reports. Although a relationship between the occurrence of SA and the nature of the stent has not been demonstrated, it is possible that stents with a higher shortening rate and lower conformability (higher axial force) may be more prone to this phenomenon. A randomized controlled trial [10] comparing the Niti-S™ D-type stent (Taewoong Medical Co., Ltd., Gimpo, Republic of Korea) with the WallFlex™ stent revealed a higher incidence of RGOO in the latter group. However, the clinical success rate did not differ between the two groups. In particular, kinking occurred in 38% of the cases and was reported as the primary cause of RGOO. While the investigators of that study did not specify the definition of kinking, it occurred at the stent end in all cases. It is presumed that this accidental symptom is identical to the SA observed in this study.

Park et al. placed duodenal stents in 318 GOO patients with unresectable gastric cancer. A total of 107 patients (33.6%) experienced SA, of whom 39 developed RGOO. Food impaction occurred in 17 patients (15.9%) in the SA group (28–486 days) and in three patients (1.4%) in the non-SA group (*p* < 0.001). In the SA group, there were two, nine, 10, and one cases of migration, overgrowth, stent collapse, and stent fracture, respectively.

In the present study, SA resulting in RGOO was observed in four patients (7%) between 2 and 32 days after duodenal stenting. All patients were able to resume oral intake after complete revision of SA by additional stent placement.

A study [18] that measured the properties of various types of colonic stents reported higher axial force in cross-wired SEMS than in hook- and cross-wired SEMS. Although the SEMS used in this study had a smaller wire diameter, abutment may have occurred due to the relatively large axial force in the cross-wired SEMS. Park et al. stated that the location of the stent end is important; if the stent is placed for pyloric stenosis, abutment may be prevented by placing the distal end of the stent longer in the duodenal D2 [17]. The WallFlex™ Soft stent used in the present study has a smaller wire diameter than the original WallFlex™ stent. However, because it is a cross-wire stent, the axial force may be higher than that of the hook-wire type. As mentioned above, all patients in whom this accidental symptom occurred experienced improvement after stent addition. Thus, unlike in the case of the hook-wire type, it may be better to select a longer stent from the beginning or to place two overlapping stents, particularly when the stent is placed in a flexure location.

In the present study, perforation was observed in two patients (3%). There was no occurrence of perforation during the procedure. The two perforations occurred at 10 and 104 days after stenting. Perforation incidence of 3% in the present study was comparable to 0–6.7% reported in a study analyzing nineteen previous prospective studies [12].

The strengths of this investigation are as follows: (1) this was a multicenter, prospective study evaluating the effectiveness of a flexible stent; (2) expansion rates over time were measured using X-ray imaging; and (3) all participating institutions exclusively used stents with a diameter of 22 mm. However, this study has the following limitations: (1) lack of a control group; (2) small sample size; and (3) the method of reintervention was selected independently by each institution, rather than uniformly.

## 5. Conclusions

In this study, a highly flexible braided stent woven with relatively thin wires was used for malignant GOO. Despite a gradual expansion with slightly lower expansile force, the stent functioned sufficiently well and showed favorable results. Nevertheless, further evaluation of the performance of this stent versus that of conventional stents is warranted.

## Figures and Tables

**Figure 1 jcm-12-00850-f001:**
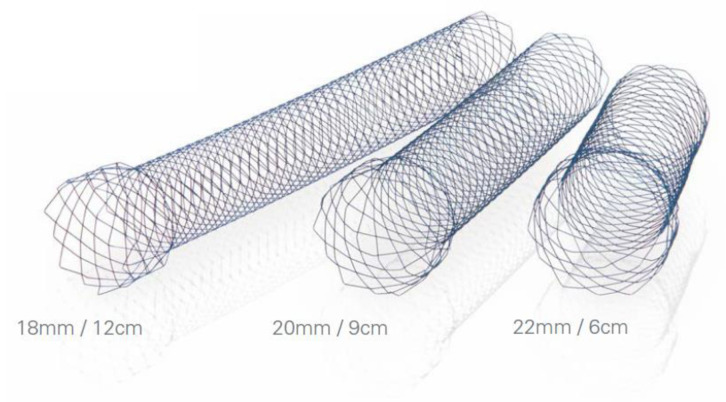
WallFlex™ Duodenal Soft.

**Figure 2 jcm-12-00850-f002:**
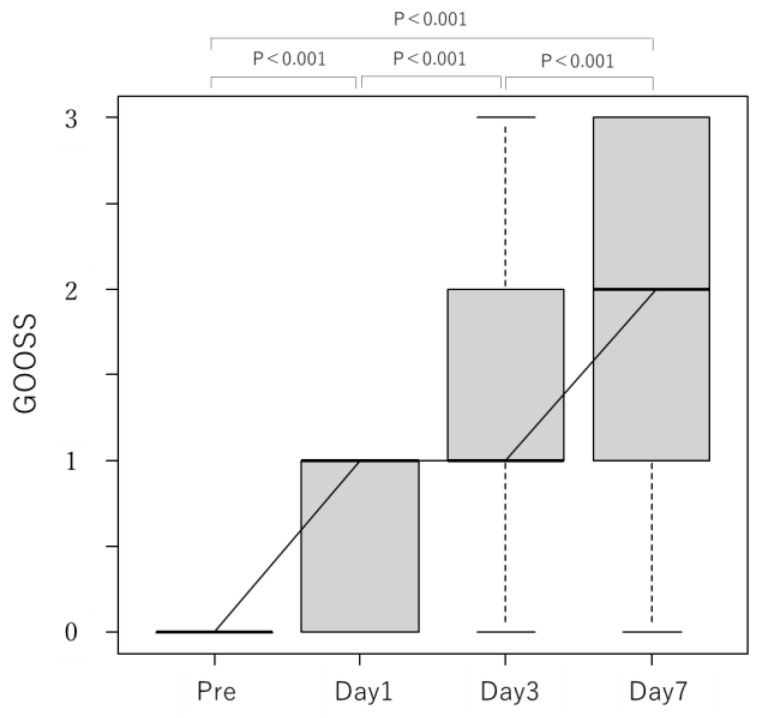
Transition of GOOSS scores up to 1 week after duodenal stent placement. The GOOSS scores prior to and at 1, 3, and 7 days after duodenal stenting were compared; these scores showed improvement after stenting. Comparisons among three or more corresponding groups (Friedman test, *p* < 0.001) showed significant differences, and Bonferroni’s multiple comparison test also showed significant differences in each group.

**Figure 3 jcm-12-00850-f003:**
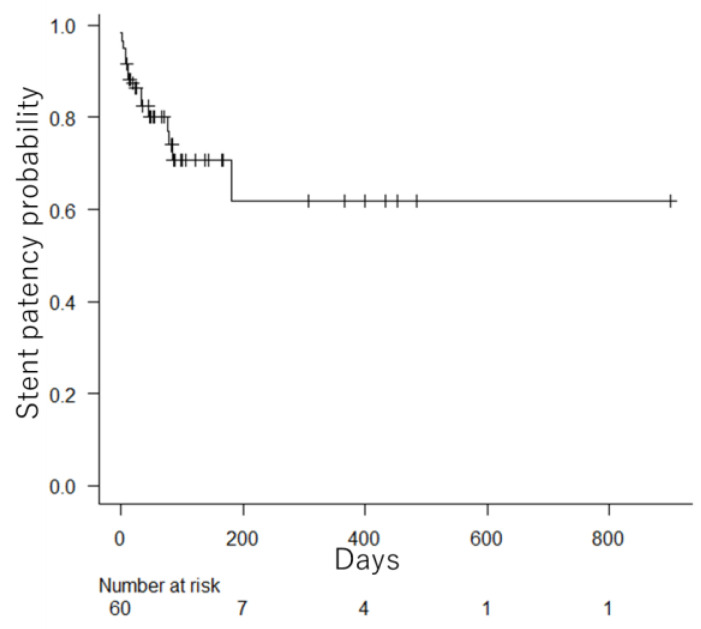
Time to RGOO (recurrent gastric outlet obstruction) after duodenal stenting. Less than half of the patients had RGOO; although there was no median, stent patency continued until death in many of those patients.

**Figure 4 jcm-12-00850-f004:**
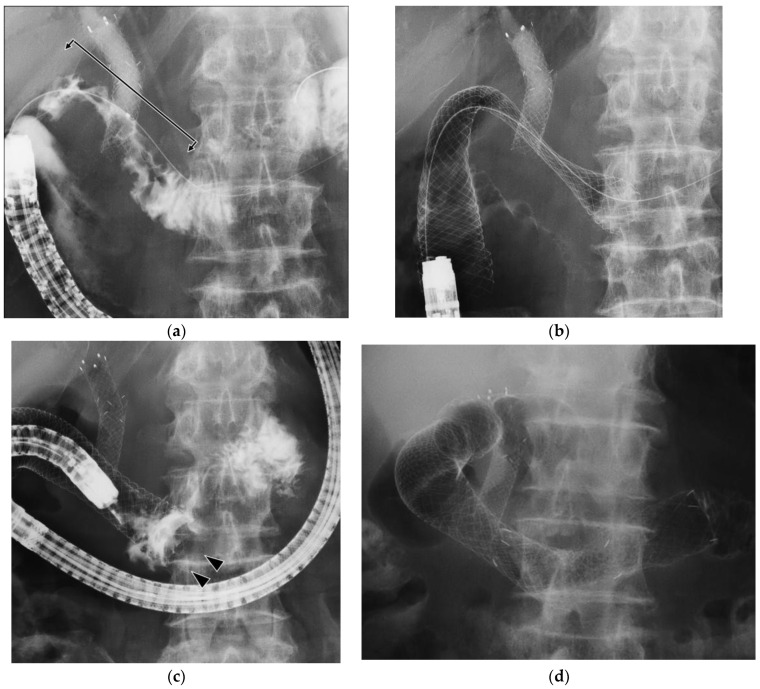
A case of abutment after duodenal stenting. (**a**): Duodenography via the endoscope showing D2 obstruction (arrow). (**b**): 12-CM stent was placed between D3 and gastric antrum. (**c**): Two days after stent placement, abutment of the anal side of the stent to the duodenal wall occurred (arrowhead), resulting in obstruction. (**d**): Additional duodenal stenting overlapping the former stent was performed for revision.

**Figure 5 jcm-12-00850-f005:**
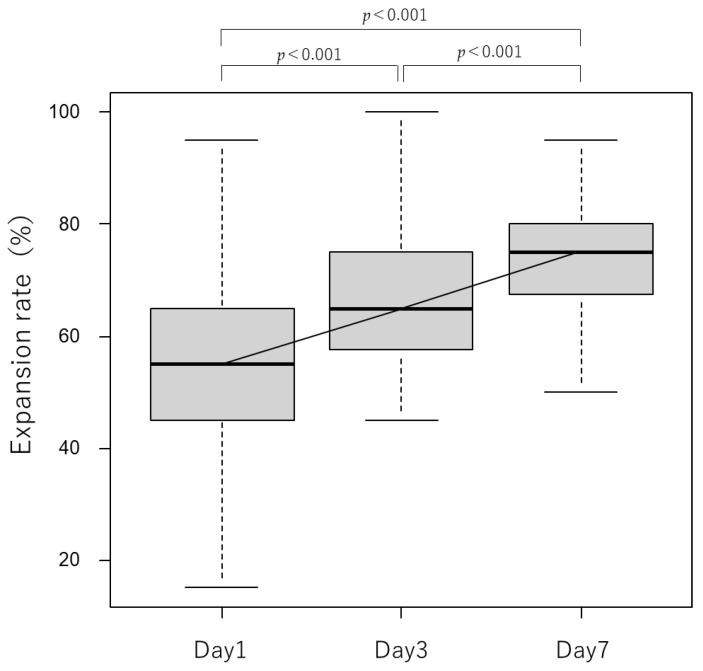
Transition of stent expansion rate. Stents were gradually expanded. Comparisons among three or more corresponding groups (Friedman test, *p*-value < 0.001) showed significant differences, and Bonferroni’s multiple comparison test also showed significant differences in each group.

**Figure 6 jcm-12-00850-f006:**
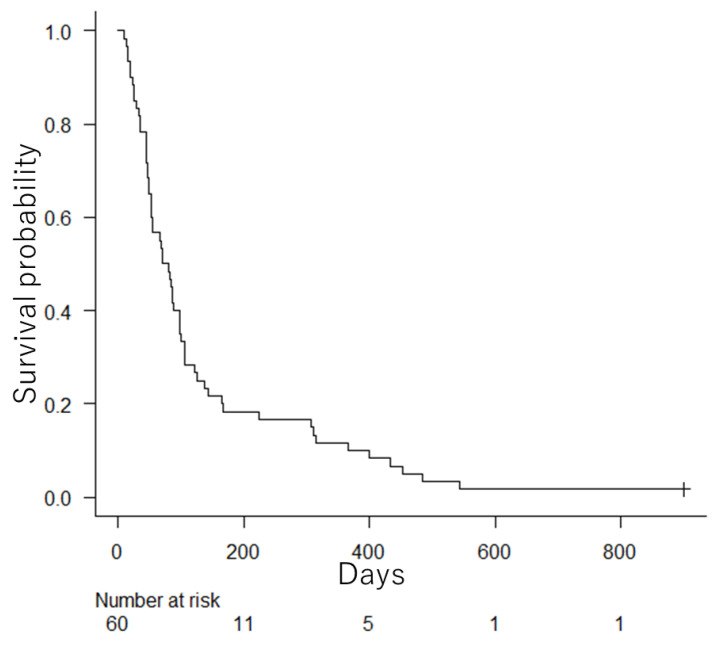
The survival curve after duodenal stenting. The median survival time was 75.5 days (95% confidence interval: 52–97 days).

**Table 1 jcm-12-00850-t001:** Characteristics of 60 patients with malignant GOO who received WallFlex™ Duodenal Soft stents (*n* = 60).

Age, years (median) [Range]	74 [40–92]
Gender (male/female)	30/30
Primary cancer site (*n*, %)	
Pancreas	32 (53)
Stomach	16 (27)
Biliary	7 (12)
Others *	5 (8)
Site of obstruction (*n*, %)	
Pylorus	10 (17)
D1	7 (12)
D2	21 (35)
D3	14 (23)
D4	2 (3)
G–D	5 (8)
G–J	1 (2)
Stricture length, cm, (median, IQR)	3 (2–4)
GOOSS (*n*, %)	
0 (no oral intake)	50 (83)
1 (liquid diet)	8 (13)
2 (soft solid diet)	2 (3)
Karnofsky performance status (*n*, %)	
100–80	22 (37)
70–50	33 (55)
40–20	5 (8)
ASA-PS (*n*, %)	
1	10 (17)
2	19 (32)
3	23 (38)
4	8 (13)
BMI (median, IQR)	19.1 (16.9–21.4)
Ascites (*n*, %)	22 (37)
Biliary intervention prior duodenal stenting (*n*, %)	
ERCP	17 (28)
PTBD	3 (5)
EUS-HGS	2 (3)
EUS-CD	0 (0)
Prior treatment (*n*, %)	
Chemotherapy	12 (20)
Radiation	1 (2)

* duodenal cancer, colorectal cancer, cancer of unknown primary, lymph node metastasis of breast cancer, and lymph node metastasis of lung cancer. D1, first part of the duodenum. D2, second part of the duodenum. D3, third part of the duodenum. D4, fourth part of the duodenum. G–D, gastroduodenal anastomosis. G–J, gastrojejunal anastomosis. GOOSS, Gastric Outlet Obstruction Scoring System. ASA-PS, American Society of Anesthesiologists physical status. BMI, body mass index. ERCP, endoscopic retrograde cholangiopancreatography. PTBD, percutaneous transhepatic biliary drainage. EUS-HGS, endoscopic ultrasound-guided hepaticogastrostomy. EUS-CD, endoscopic ultrasound-guided choledochoduodenostomy.

**Table 2 jcm-12-00850-t002:** Procedure of duodenal stenting.

Number of SEMS (*n*, %)	
Single	57 (95)
Double	3 (5)
Length of SEMS, cm (*n*, %)	
6	9 (15)
9	26 (43)
12	22 (37)
12 + 6	1 (2)
12 + 9	2 (3)
Procedure time, min (median, IQR)	20 (15–30)
Endoscope used (*n*, %)	
Forward viewing	35 (58)
Side viewing	25 (42)
Simultaneous biliary drainage (*n*, %)	
ERCP	6 (10)
PTBD	2 (3)
EUS-HGS	4 (7)
EUS-CD	1 (2)

SEMS, self-expandable metallic stent. ERCP, endoscopic retrograde cholangiopancreatography. PTBD, percutaneous transhepatic biliary drainage. EUS-HGS, Endoscopic ultrasound-guided hepaticogastrostomy. EUS-CD, Endoscopic ultrasound-guided choledochoduodenostomy.

**Table 3 jcm-12-00850-t003:** Clinical outcomes in patients after stent placement.

Technical success (*n*, %)	60 (100)
Clinical success (*n*, %)	54 (90)
GOOSS after procedure (*n*, %)	
Median (IQR)	2 (1–3)
0 (no oral intake)	4 (7)
1 (liquid diet)	13 (20)
2 (soft solid diet)	15 (25)
3 (low residue or normal diet)	28 (47)
Improvement of GOOSS, (median, IQR)	2 (1–3)
Procedure-related adverse event (*n*, %)	
Cholangitis	3 (5)
Perforation	2 * (3)
Cholecystitis	1 (2)
Pancreatitis	1 (2)
Bleeding	1 * (2)
Additional treatment after stent placement (*n*, %)	
Chemotherapy	12 (20)
Radiation	1 (2)
Median time of follow-up, days (IQR)	75.5 (44–128)
RGOO (total) (*n*, %)	15 (25)
Stent in growth	7 (12)
Abutment	4 (7)
Insufficient expansion	3 ** (5)
Stent migration	2 ** (3)
Stent kinking	0
Food impaction	0
Additional stent placement (*n*, %)	13 (22)
Stent expansion rate, median (IQR)	75 (68–80)

* one case with both perforation and bleeding. ** one case with both insufficient expansion and stent migration. GOOSS, Gastric Outlet Obstruction Scoring System. RGOO, recurrent gastric outlet obstruction.

## Data Availability

The data generated during this study are available within the article. Datasets analyzed during the current study preparation are available from the corresponding author on reasonable request.

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
