# Peer review of "Gastroduodenal Stenting with a Flexible Stent Demonstrates Favorable Clinical Effectiveness despite Gradual Expansion: A Multicenter Prospective Study"

_jcm, 2023, doi:10.3390/jcm12030850_

Round 1
Reviewer 1 Report
Great paper highlighting a common issue of inadequate duodenal opening after stent placement.
In methods section, would suggest explaining briefly GOOSS and scoring system as it would be easier for readers in general and would be important to understand results and discussion later.
Also, the study does not have a control group
Few minor comments-
1. Line 40, the authors mentioned “Moreover, the procedure may not be effective in numerous cases”. Would suggest rephrasing the sentence as “numerous” is a vague term.
2. In the stenting procedure section, can authors comment on expertise or duration of training of the endoscopist or proceduralists who performed this procedure. Was there any additional training provided to specifically to perform this procedure.
3. Page 4, line 131, can authors name the other diseases in which this procedure was used either in text or in table
4. In Discussion section, improvement of GOOSS score by at least 1 point is explained in paragraph 3 as well as 4. May be this information can be removed refer to lines 269-271
5. Line 301, RDO can be expanded. If authors mean RGOO then can be corrected
6. Conclusion – line can be rephrased “Despite a slightly slower expansion “versus” stent with higher radial force, the stent functioned sufficiently well and showed favorable results.
Author Response
In methods section, would suggest explaining briefly GOOSS and scoring system as it would be easier for readers in general and would be important to understand results and discussion later.
We inserted in 2 Methods “GOOSS was used to evaluate the severity of obstructive symptoms (GOOSS 0 = no oral intake, 1 = liquids, 2 = soft solids, 3 = low residue or full diet), proposed by Adler and Baron.”
We also added it to the reference. “Adler, DG.; Baron, TH., Am. J. Gastroenterol 2002, 97, 72–78.”
Few minor comments-
- Line 40, the authors mentioned “Moreover, the procedure may not be effective in numerous cases”. Would suggest rephrasing the sentence as “numerous” is a vague term.
We rephrased “in about 10% of cases”.
- In the stenting procedure section, can authors comment on expertise or duration of training of the endoscopist or proceduralists who performed this procedure. Was there any additional training provided to specifically to perform this procedure.
We added “All procedures were carried out by expert endoscopists in each hospitals being familiar with gastroduodenal stenting.”
- Page 4, line 131, can authors name the other diseases in which this procedure was used either in text or in table
We added it to the Table annotation. “duodenal cancer, colorectal cancer, cancer of unknown primary, lymph node metastasis of breast cancer, and lymph node metastasis of lung cancer.”
- In Discussion section, improvement of GOOSS score by at least 1 point is explained in paragraph 3 as well as 4. May be this information can be removed refer to lines 269-271
We changed the original into “clinical success was achieved in 90% of enrolled cases”.
- Line 301, RDO can be expanded. If authors mean RGOO then can be corrected
We corrected “RDO” to “RGOO”.
- Conclusion – line can be rephrased “Despite a slightly slower expansion “versus” stent with higher radial force, the stent functioned sufficiently well and showed favorable results.
We rephrased “Despite a gradual expansion with slightly lower expansile force, the stent functioned sufficiently well and showed favorable results.”
Reviewer 2 Report
The present observational prospective study analyses the role of a new enteral stent thought to be more effective and flexible in challenging stenosis. The study is very interesting and focused on a new device, however, the definitions used by the authors are too flexible and have probably "improved" the results of the study. Notably, a higher proportion of patients with "clinical success" did not achieve a solid diet!
There is a higher number of patients with distal stenosis. Distal stenosis are more difficult to treat and recurrent GOO is higher. It would be helpful for the readers to exaplein this fact and deeply analyse the results regarding the site/location of the stenosis.
It's difficult to understantd why several stents were placed in 3 patients if the technical success was 100%? A re-stenting during the same procedure is usually done because the first stent was not enough or technically successfull, these patients should probably be considered as technical failure, 1 stent per patient....
Major points:
The definition of clinical success seems not really accurate and different from previously published studies. The authors have defined clinical success as the improvement in only 1 point in the GOOSS score. Similarly, stent obstruction has been defined as GOOSS score of 0 but a score of 1 is also considered as an obstruction (no solid diet) in many papers and an indication of stenting. These definitions have probably improved the overall results of the study and can lead to a bias. In addition, please be more clear in the remaining definitions, authors cannot define "inssuficient expansion" as "no expansion" using the same word/term...
Author Response
The present observational prospective study analyses the role of a new enteral stent thought to be more effective and flexible in challenging stenosis. The study is very interesting and focused on a new device, however, the definitions used by the authors are too flexible and have probably "improved" the results of the study. Notably, a higher proportion of patients with "clinical success" did not achieve a solid diet!
There is a higher number of patients with distal stenosis. Distal stenosis is more difficult to treat and recurrent GOO is higher. It would be helpful for the readers to explain this fact and deeply analyze the results regarding the site/location of the stenosis.
It's difficult to understand why several stents were placed in 3 patients if the technical success was 100%? A re-stenting during the same procedure is usually done because the first stent was not enough or technically successful, these patients should probably be considered as technical failure, 1 stent per patient....
Two overlapping stents were intentionally performed for long obstruction in these three patients. Thus, these cases were considered as technical success.
Major points:
The definition of clinical success seems not really accurate and different from previously published studies. The authors have defined clinical success as the improvement in only 1 point in the GOOSS score. Similarly, stent obstruction has been defined as GOOSS score of 0 but a score of 1 is also considered as an obstruction (no solid diet) in many papers and an indication of stenting. These definitions have probably improved the overall results of the study and can lead to a bias.
The definition of clinical success of duodenal stenting varies among studies, without any consensus. Like the present study, some previous studies [Ref#7, 12] defined clinical success as an improvement of at least 1 point of GOOSS regardless of possibility of solid diet.
In addition, please be more clear in the remaining definitions, authors cannot define "insufficient expansion" as "no expansion" using the same word/term...
We rephrased to line 116, ”Insufficient expansion was defined as unsatisfactory expansion necessitates balloon dilation and/or additional stent placement.”
Round 2
Reviewer 2 Report
I have no more comments